# Novel Simple Approach for Production of Elastic Poly(propylene carbonate)

**DOI:** 10.3390/polym16233248

**Published:** 2024-11-22

**Authors:** Elena S. Trofimchuk, Igor V. Chernov, Roman V. Toms, Sergey A. Rzhevskiy, Andrey F. Asachenko, Anna V. Plutalova, George A. Shandryuk, Elena V. Chernikova, Irina P. Beletskaya

**Affiliations:** 1Faculty of Chemistry, Lomonosov Moscow State University, Lenin Hills, 1, Bld. 3, 119991 Moscow, Russia; civ-100@mail.ru (I.V.C.); annaplutalova@gmail.com (A.V.P.); beletska@org.chem.msu.ru (I.P.B.); 2Faculty of Materials Science, Shenzhen MSU-BIT University, Longgang District, Shenzhen 518172, China; 3Institute of Fine Chemical Technologies named by M.V. Lomonosov, MIREA—Russian Technological University, pr. Vernadskogo, 86, 119571 Moscow, Russia; toms.roman@gmail.com; 4Topchiev Institute of Petrochemical Synthesis, Russian Academy of Sciences, Leninskiy av., 29, 119991 Moscow, Russia; rs89a@yandex.ru (S.A.R.); aasachenko@ips.ac.ru (A.F.A.); gosha@ips.ac.ru (G.A.S.)

**Keywords:** poly(propylene carbonate), ring-opening copolymerization, cyclodepolymerization, elastic properties, polymer films

## Abstract

The simple approach of increasing the elastic properties of atactic poly(propylene carbonate) (PPC) with Mn = 71.4 kDa, ĐM = M_w_/M_n_ = 1.86, and predominantly carbonate units (>99%) is suggested by selecting the appropriate hot pressing temperature for PPC between 110 and 140 °C. Atactic PPC is synthesized through ring-opening copolymerization of (*rac*)-propylene oxide and CO_2_ mediated by racemic salen complex of Co(III). Hot pressing PPC results in the release of a small amount of propylene carbonate (PC), sufficient to lower the glass transition temperature from 39.4 to 26.1 °C. Consequently, increasing the pressing temperature from 110 to 140 °C generates materials with a reduced modulus of elasticity (from 1.94 to 0.09 GPa), yield strength (from 38 to 2 MPa) and increased tensile elongation (from 140 to 940%). Thermomechanical analysis has shown a significant expansion in sample volume by hundreds of percent within the 80–130 °C range. PPC also displays large, reversible deformations, which can be utilized by creating shape memory materials.

## 1. Introduction

The design of chemical products and processes that reduce or eliminate the use and generation of hazardous substances lies at the heart of Green Chemistry. From this viewpoint, replacing traditional plastics with biodegradable plastics in areas where the lifespan of polymer materials is short, such as in packaging, aligns with the principals of Green Chemistry [1]. Biodegradable plastics should contain weak bonds capable of scission under the influence of the environment (microorganisms, oxidation, temperature, etc.) [2]. Polycarbonates which contain –O–C(=O)–O– moiety in each monomer unit are a typical example of biodegradable polymers [3]. Polycarbonates can be synthesized either through step polymerization (polycondensation) [4] or through chain polymerization [5,6,7,8,9,10,11,12,13,14,15,16,17]. However, only the latter process can be considered atom-effective or atom-economical and in line with the principles of Green Chemistry. The production of polycarbonates through ring-opening copolymerization (ROCP) involves epoxides, carbon dioxide, and a catalyst. Since their invention by Inoue [18], polycarbonates have remained attractive and interesting for scientists and industry [19]. The modern trends in this field deal with the development of metal-free catalysts, the use of bio-based monomers and the improvement of the properties of known polycarbonates [11,20,21,22,23,24,25,26,27,28,29,30,31,32,33,34,35,36,37,38,39].

Poly(propylene carbonate) (PPC) is the most studied among other polycarbonates and is the first aliphatic polycarbonate produced on an industrial scale [40,41,42,43,44,45,46,47]. The presence of an asymmetric carbon atom in propylene oxide, along with the choice of a suitable catalyst, allows for the production of polypropylene carbonate with different microstructures, such as stereoregular (iso- and syndiotactic or stereogradient) and irregular (atactic), with varying ratios of head-to-tail and head-to-head linkages [40]. PPC may also contain some amount of ether groups depending on polymerization conditions [48,49,50]. Therefore, the variety of possible configurations, the presence of ether groups, different end groups, and impurities from catalysts, among other factors, contribute to a wide range of mechanical properties and thermal stability in PPC [3,23,24,25,51].

Isotactic and stereogradient PPC exhibit high onset degradation at 240 and 273 °C, respectively [52], while irregular (atactic) PPC has lower thermal stability ranging from 180 to 220 °C [15]. The thermal stability of PPC is decreased by the presence of catalysts, bases, and water [8]. However, if the terminal hydroxyl groups of the polymer are protected by end capping, the thermal stability of PPC increases [53].

There is a lack of information about mechanical properties of stereoregular PPC, which should be capable of crystallizing. In contrast, atactic PPC, as an amorphous polymer, can have a glass transition temperature (T_g_) in the range of 8 to 45 °C depending on the content of carbonate units in the chain [8,51,54,55]. The presence of ether units in the macromolecule increases flexibility and lowers the T_g_. Thus, lower values of T_g_ are observed for PPC synthesized in the presence of some heterogeneous catalysts such as Zn dicarboxylates, which are unable to provide high chemoselectivity of the ROCP [47]. The higher T_g_ values are observed for PPC synthesized by using homogeneous catalysts such as salen complexes of cobalt, which are characterized by high chemoselectivity [55]. The consequence of the low glass transition temperature of PPC is its low strength characteristics. PPC with M_n_ ranging from 70 to 260 kDa and Đ_M_ = 3.2–5.0 has a Young’s modulus of 680 MPa and a tensile strength of 5 to 17 MPa [15,38]. PPC with a M_n_ = 50 kDa and Đ_M_ of 1.07 has a Young’s modulus of 830 MPa, breaking elongation of 330%, and a tensile strength of 21.5 MPa [15].

Mechanical properties can also be adjusted by copolymerization of propylene oxide (PO)/CO_2_ with other epoxides or anhydrides [3], by blending PPC with other polymers, by the use of additives, fillers/reinforcing agents [56], or by plasticizing PPC [42]. Introducing ethylene oxide or butylene oxide reduces T_g_, while copolymerization with bulky epoxides, e.g., styrene oxide, cyclohexane oxide, limonene oxide, etc., results in the increase in T_g_ of resulting polycarbonate [42,57,58]. The latter effect is also achieved when aromatic anhydrides are copolymerized with PO [47]. However, the listed approaches have some drawbacks. Firstly, finding an efficient and cost-effective catalyst for terpolymerization of propylene oxide (PO) and another epoxide with CO_2_ is challenging, as different catalysts are typically used for these epoxides. Secondly, the formation of ether units during copolymerization of PO and CO_2_ indicates the low chemoselectivity, which may not be reproducible. Thirdly, blending with a thermodynamically incompatible polymer results in macrophase segregation of polymers. Finally, using a plasticizer requires an additional step in the technological process, namely, mixing the plasticizer with the polymer.

In the present research, we propose a new simple approach that allows for the soft tuning of elastic properties of atactic PPC, which predominantly contains carbonate units (>99%). The idea is based on PPC’s ability to release propylene carbonate (PC) through cyclodepolymerization. PC is a non-volatile (b.p. 250°C), and it has high compatibility with PPC. Therefore, it can plasticize a polymer, decreasing its glass transition temperature and enhancing the elastic properties of the polymer. The controlled release of a small amount of PC (3–6 wt. %) is achieved by hot pressing PPC for a relatively short period at the desired temperature much below the upper limit temperature of the polymerization.

## 2. Materials and Methods

### 2.1. Materials

All air- or water-sensitive reactions were carried out under dry nitrogen using either a drybox or standard Schlenk-line techniques. Methylene chloride (“Component-reaktiv”, Russia; “dry”) and diethyl ether (“Component-reaktiv”, Russia; “reagent grade”) were dried and degassed by passing them through a column of activated alumina and sparging with dry nitrogen. (*rac*)-1,2-Diaminocyclohexane (Sigma Aldrich, St. Louis, MO, USA, 99%) and 3,5-di-*tert*-butylsalicylaldehyde (Sigma Aldrich, St. Louis, MO, USA, 99%) were used as received. PPNCl was prepared following a literature procedure [59]. (*Rac*)-propylene oxide (Sigma Aldrich, St. Louis, MO, USA, ≥99%) was dried over calcium hydride, distilled under an argon atmosphere and vacuum transferred before use.

### 2.2. Synthetic Procedures

#### 2.2.1. Synthesis of the Complex Rac-(salcy)Co(III)OBzF_5_

*rac*-(Salcy)Co(III)OBzF_5_ (where OBzF_5_ means OC(=O)C_6_F_5_) was synthesized using a modified procedure involving a series of transformations. The process began with (rac)-1,2-diaminocyclohexane and 3,5-di-t-butylsalicylaldehyde, which were used to create (rac)-salcy ligand. This ligand was then used to produce (rac)-(salcy)Zn, followed by (rac)-(salcy)Co(II), and finally (rac)-(salcy)Co(III)OBzF_5_ [60,61].

#### 2.2.2. Polymer Synthesis

The mixture of the (*rac*)-(salcy)CoOBzF_5_ complex (0.014 mmol) and PPNCl (0.016 mmol) was dissolved in (*rac*)-PO (6 mL, 85.7 mmol, 6000 equiv.) in a 10 mL vial equipped with a teflon-coated magnetic stir bar. The mixture was allowed to stir until a red-brown homogeneous solution was formed. Then, the vial was placed into a pre-dried 100 mL autoclave, which was pressurized to the appropriate pressure with CO_2_. After 64 h, the mixture was dissolved in 50 mL of CH_2_Cl_2_ and concentrated under vacuum. The polymer was isolated by precipitation from CH_2_Cl_2_/MeOH (10/1, *v*/*v*) solution in tenfold excess of diethyl ether, followed by filtration and long-term drying in vacuum.

### 2.3. Instrumentation

NMR spectra were obtained on a Bruker Avance III HD (400 MHz ^1^H, 101 MHz ^13^C). The chemical shifts are frequency referenced relative to the residual undeuterated solvent peaks.

The SEC measurements were performed in THF at 40 °C with a flow rate of 1.0 mL/min using a 1260 Infinity II GPC/SEC Multidetector System chromatograph (Agilent, Santa Clara, CA, USA) equipped with two PLgel 5 μm MIXED B columns. The SEC system was calibrated using narrow dispersed linear poly(methyl methacrylate) standards with MW ranging from 0.8 to 2000 kDa.

Differential scanning calorimetry measurements were conducted using a Netzsch DSC 204 (Netzsch, Germany) and a Mettler Toledo TGA/DSC3+ system (Mettler Toledo, Greifensee, Switzerland) using 40 μL Al pans at a heating rate of 10 °C min^−1^ under an argon atmosphere. Melting points and enthalpies of indium and zinc were used for temperature and heat capacity calibration.

Thermogravimetric analysis was performed using Mettler TA4000 system and a Mettler Toledo TGA/DSC3+ system at a heating rate 10 °C min^−1^ under a nitrogen atmosphere.

PPC films with a thickness of 100 μm were produced by hot pressing using a custom-designed thermal press at temperatures of 110, 130, and 140 °C and a pressure of 18 MPa for 10 min. Afterwards, the films were cooled using an air flow at a temperature of 20 °C.

Thermomechanical properties were studied using Mettler Toledo TMA/SDTA 2+. The data were processed using STARe v.16.20 (Mettler Toledo) service program.

The mechanical behavior of the PPC films was investigated using a stress–strain test in uniaxial stretching mode before rupture on a universal testing machine, the Z3-X500 from Thümler, Germany, equipped with a 50 N load cell and Nordic Transducer, Denmark, under air conditions at speeds of 2, 10, and 50 mm min^−1^ at 24 °C. The samples were also tested in cyclic tensile loading–unloading mode at a speed of 1 mm min^−1^. Before testing, the samples were cut from the films using a special knife with a double-sided blade, with a working blade length of 10 mm and a width of 5.2 mm.

## 3. Results and Discussion

### 3.1. Polymer Synthesis and Characterization

To synthesize amorphous PPC, a racemic epoxide and catalyst were used [41]. Copolymerization of racemic PO and CO_2_ was carried out using (*rac*)-(salcy)CoOBzF_5_ (salcy = *N*,*N*′-bis(3,5-di-*tert*-butylsalicylidene)-1,2-diaminocyclohexane) catalyst (Cat) and a bis(triphenylphosphine)iminium chloride (PPNCl) co-catalyst (co-Cat) in a molar ratio of [PO]/[Cat]/[co-Cat] = 6400/1/1.16 with an initial CO_2_ pressure of 2.5 MPa and 21 °C during 64 h. Recently, we have conducted the detailed study of the kinetics and mechanism in PO/CO_2_/Cat/co-Cat system [62]. We have demonstrated that copolymerization proceeds via a living mechanism, which is violated at high conversions due to the increasing viscosity of the system. The ligand exchanged between Cat and co-Cat plays an important role in the polymerization kinetics, and the general scheme of the polymerization mechanism can be presented as follows (Figure 1).

The conversion of PO was 78.1%, with TON = 5046 and TOF = 78.8 h^−1^. The synthesized polymer has M_n_ = 71.4 kDa and Đ_M_ = 1.86 (Figure 1, curve 1).

The chemical composition and microstructure of the purified polymer product were confirmed by ^1^H and ^13^C NMR spectroscopy (Figure 2). The polymer consisted mainly of carbonate units (4.98 (CH), 4.27–4.10 (CH_2_), and 1.31 ppm (CH_3_)), with no ether units detected at 3.56 (CH), 3.38 (CH_2_), and 1.12 ppm (CH_3_) (Figure 2a). The formation of head-to-tail (HT) regioisomers with relatively small proportions of head-to-head (HH) and tail-to-tail (TT) linkages was predominant with a molar ratio of the units HT:TT:HH = 6.4:1.0:0.9 (Figure 2b). The racemic cobalt complex led to the formation of an atactic polymer configuration (mr/rm > rr ≈ mm) during the copolymerization of racemic PO and CO_2_. Therefore, PPC was expected to exhibit the typical behavior of an amorphous polymer.

The thermal stability of PPC sets the upper limit for the temperature range in which it can be processed and operated. For PPC with uncapped terminal groups, thermal stability is relatively low [3]. The temperature of 5% weight loss of pristine PPC is 160 °C, and 50% weight loss occurs at 194 °C (Figure 3a). DSC shows a noticeable endo-effect in the similar temperature range corresponding to the degradation process (Figure 3b). The activation energy (E_a_) of PPC thermal degradation was estimated from DSC (Figure 4) and TGA analysis (Figure 5) at different heating rates using the Kissinger method [63]:(1)$$-\frac{E_{a}}{R}=\frac{d\left[ln\left(\frac{\varphi }{T_{p}^{2}}\right)\right]}{d\left(\frac{1}{T_{p}}\right)}$$
where *R*—universal gas constant; *T_p_*—peak temperature (K); *φ*—heating rate (K min^−1^).

The increase in the heating rate results in the shift of thermograms to higher temperatures and an increase in the intensity of heat flow. Both pristine PPC and PPC subjected to hot pressing at 130 °C exhibit similar behavior (Figure 4a,b). A Kissinger plot (Equation (1)) was used to determine the activation energy of PPC degradation (Figure 4c). Both polymers have similar values of activation energy of degradation, 69.1 ± 7.5 and 79.3 ± 13.0 kJ mole^−1^, respectively. DTG analysis of pristine PPC (Figure 4d) gives the value of activation energy of PPC degradation equal to 60.7 ± 2.7 kJ mole^−1^. This data spread is acceptable when comparing the results obtained using different methods. Overall, it can be concluded that the heat treatment of PPC does not significantly affect its ability to degrade. 

These values differ slightly from those described in the literature [8,64,65,66]. The value of E_a_ was found to be sensitive to experimental conditions and the molecular characteristics of PPC. The value of E_a_ for thermal degradation PPC with M_n_ = 130 kDa and Đ_M_ = 3.7 synthesized using rare earth complex was equal to 99.9 kJ mole^−1^ [64]. For PPC that underwent melt processing in a twin-screw extruder, the E_a_ was about 110–120 kJ mole^−1^ depending on the rotation rate of the extruder [65]. The activation energy of thermal degradation of PPC in the presence of salen complex of Cr and Bu_4_NN_3_ was c.a. 80 kJ mole^−1^ [66].

Thus, the processing temperature for the synthesized PPC should be lower than 160 °C to avoid noticeable degradation of the polymer. Hot pressing of PPC at temperatures much below the degradation temperature results in the production of transparent polymer films with different levels of stiffness. The most rigid sample was produced by pressing PPC at 110 °C. Increasing the pressing temperature resulted in softer materials that were more flexible and easily deformed by hand. According to DSC, pristine PPC and the films have different glass transition temperatures (Figure 5, Table 1).

One can say that the change of T_g_ may originate from a decrease in MW due to the partial degradation of PPC during thermal treatment. To investigate this, we analyzed the MWD of PPC before and after hot pressing (Figure 1). As shown, a degradation process occurs leading to a decrease in the MW of the polymer, which is more significant at higher temperatures. However, even after PPC was pressed at 140 °C (Table 1), it still had a sufficiently high MW, exceeding the entanglement molecular weight (M_e_) of the PPC by several times, which is 17 kDa [51]. Therefore, the decrease in MW is not the cause of the decrease in T_g_. This conclusion can be supported by the following data. We have synthesized PPC with M_n_ = 77.4 and Đ_M_ = 1.30 with T_g_ 36.4 °C and PPC with M_n_ = 52.1 and Đ_M_ = 1.76 with T_g_ 37.4 °C. Additionally, PPC3 was precipitated from chloroform solution in methanol and dried; T_g_ of the dried PPC3 increased to 37.2 °C. Hence, MW (when MW exceeds M_e_) and MWD have no visible effect on T_g_.

Then, we assume that the decrease in glass transition temperature is caused by the release of a certain amount of PC. PC is characterized by a high boiling point (above 250 °C), making its evaporation during hot pressing unlikely. The film obtained after pressing at 130 °C was dissolved in CDCl_3_ and analyzed by ^1^H NMR (Figure 6). According to ^1^H NMR spectroscopy data, the chemical structure of PPC does not change as the number of carbonate units in the macromolecules remains constant. However, the low-intensity signals corresponding to propylene carbonate can easily be distinguished in the spectrum; ^1^H NMR: δ 4.84 (CH), 4.54 (CH_2_), 4.01 (CH_2_), 1.48 ppm (CH_3_). The molar content of PC in the sample is 3.4%. Therefore, PC is formed during hot pressing in the polymer matrix, and it can serve as a plasticizer lowering the glass transition temperature of PPC. This conclusion is supported by data on the influence of PC on the T_g_ of PPC [51]. The analysis of these data allows us to estimate the change in T_g_ depending on the volume fraction ϕ_PC_ as ΔT_g_ ≈ 192ϕ_PC_. The observed decrease in T_g_ corresponds to the 3–6 vol.% of PC, which aligns to NMR data.

Thus, we have discovered that the formation of PC occurs during hot pressing at temperatures much below the degradation of the polymer. It is evident that the presence of a plasticizer should affect the mechanical characteristics of the PPC films and allow for the production of materials with different properties.

### 3.2. Mechanical Properties of PPC

#### 3.2.1. Thermomechanical Analysis

The pristine PPC exhibited noticeable plasticity at room temperature. Its thermomechanical properties were first studied under constant loading conditions at 0.1 N. As a result, three temperature transitions were identified, where the mechanical properties of the polymer significantly changed (Figure 7). The first transition occurred in the temperature range of 30–40 °C, where the polymer’s ductility noticeably increased. This transition was consistent with the findings from DSC studies and was identified as the glass transition (insert in Figure 7). An intensive increase in the volume of the sample by tens or even hundreds of percent was observed above 90 °C. It is mentioned in the literature that, unlike other polymers, PPC exhibits a very high thermal expansion of 3699.9 μm m^−1^K^−1^ [67], which exceeds the values of most known polymers by ca. 30 times. Above 160 °C, a loss of sample thickness began, with complete loss observed above 200 °C, likely due to the thermal degradation (cyclodepolymerization) of PPC. This observation aligns with the TGA and DSC data.

In order to study the reversibility of the mechanical properties of the material above glass transition temperature, we conducted experiments on the cyclic heating and cooling of samples before any noticeable fluidity occurred. In the first experiment, the sample was heated at a rate of 2 °C min^−1^ from room temperature up to 110 °C, resulting in a 6.7% change in the thickness of the sample and then cooled at the same rate back to room temperature (Figure 8a). It can be seen that after cooling, the initial thickness of the sample was almost completely restored. This fact aligns with the observation by Lee et al. [68] that cooling the PPC after molding results in shrinkage to dimensions close to the initial values. However, when the sample was heated to a higher temperature of 130 °C (resulting in a 30% change in thickness), complete restoration did not occur (Figure 8b).

#### 3.2.2. Stress–Strain Test in Uniaxial Stretching Mode

In the mechanical tests, the PPC films that had been subjected to hot pressing were used. The tests were carried out at room temperature. A clear difference between the PPC films that underwent hot pressing at different temperatures was demonstrated. Figure 9 illustrates the stress–strain curves for uniaxial stretching of films that were obtained by hot pressing at temperatures of 110, 130, and 140 °C, at different stretching rates. Uniaxial stretching of films pressed at 110 °C occurs through the formation of a well-defined local constriction on the working part and the gradual expansion of the neck, which is typical of the deformation of glassy polymers. The graph for this sample has a classical appearance: an initial linear section (the region of elastic behavior), a yield point in the form of a maximum, and then a plateau that shows the propagation of the neck. An increase in the stretching rate results in a slight increase in Young’s modulus, a significant increase in yield strength, and a decrease in tensile elongation (Figure 9a, Table 2).

For glassy polymers, a linear dependence should be observed within the frameworks of Eyring rate theory [69]:$$\frac{\sigma_{y}}{T}=\frac{U}{aT}+\frac{k}{a}\times \frac{lnV}{A},$$
where σ_y_ is a yield point value; *U*—activation energy; *a*—the coefficient having the dimension of volume proportional to the volume of the segment; *k*—the Boltzmann constant; *V*—stretching rate; *T*—temperature; A—pre-exponential factor. The corresponding dependence of the Eyring plot (Figure 10) for PPC film pressed at 110 °C linearly confirms the glassy behavior of PPC.

For films produced at higher temperatures, these dependences generally persist but are less pronounced. Films made at 130 °C exhibit a less noticeable neck with a limit of 19 MPa at a stretching rate of 10 mm min^−1^, although the overall curve shape remains (Figure 9b). In the case of a PPC film pressed at 140 °C, no neck forms when deformed at a rate of 2 and 10 mm min^−1^, and the deformation occurs uniformly (Figure 9c). For a sample deformed at a rate of 10 mm min^−1^, the yield point is weakly expressed at 2 MPa, and stress increases linearly from 1.5 to 3.5 MPa over a wide range of deformations (from 200 to 900%) (Table 2). These low tensile stress values are typically seen in polymers just above their glass transition temperature.

Therefore, an increase in the pressing temperature results in the production of materials with a lower modulus of elasticity, which decreases by up to 20 times. Additionally, there is a decrease in yield strength by more than 15 times and an increase in tensile elongation by 6 to 940%. These changes in the mechanical behavior of PPC are linked to a gradual decrease in the glass transition temperature of the polymer caused by the formation of propylene carbonate, which acts as a good plasticizer for PPC.

#### 3.2.3. Cyclic Tensile Loading–Unloading Mode (Hysteresis)

The reversibility of large deformations in PPC films has been investigated through a cyclic tensile loading−unloading test at room temperature (Figure 11). For a film prepared at 110 °C, which is in a glassy state, the ability to recover from deformations during the test is low (Figure 11a). The proportion of recoverable deformation does not exceed 20–30%. There is also a significant reduction in stress during repeated deformations, which is associated with the phenomenon known as force softening in polymer materials. However, a longer relaxation of the sample (within a week) in an unloaded state at a temperature of 23–25 °C after stretching to 200% resulted in a significant restoration of its working length. The value of relative shrinkage was 80%.

Additionally, the mechanical properties of the film were significantly restored due to the relatively low glass transition temperature of the polymer, its proximity to the testing temperature, and the presence of a relatively good physical mesh of engagement.

In the case of cyclic loading at room temperature of the PPC film obtained at 130 °C, the reversibility of deformation is higher and occurs at higher rates (Figure 11b). Thus, after stretching the film by 500%, the relative shrinkage reached 75%. After relaxing in the free state for 18 h, the sample had almost returned to its original size with shrinkage exceeding 95%. These results indicate the absence of flow processes leading to the accumulation of irreversible deformations during the stretching of PPC at relatively low rates.

In summary, the temperature of hot pressing determines the amount of the released PC and the level of stiffness of the polymer films tested at room temperature. An increase in the temperature of hot pressing makes the film more elastic and accelerates shrinkage after unloading the sample. An increase in the stretching rate results in a slight increase in Young’s modulus, a significant increase in yield strength, and a decrease in tensile elongation, due to the decrease in the time for the mechanical response of the polymer, which accords with the principle of time–temperature superposition.

## 4. Conclusions

In this research, we conducted a comprehensive analysis of the mechanical characteristics of PPC films obtained by hot pressing at various temperatures. We have shown that an increase in the pressing temperature from 110 to 140 °C leads to a decrease in the T_g_ of the polymer due to the release of propylene carbonate. The amount of propylene carbonate released depends on the temperature of hot pressing of the polymer. Propylene carbonate being a good plasticizer for PPC allows the fine-tuning of the mechanical characteristics of PPC. This results in a decrease in Young’s modulus by up to 20 times, yield strength by more than 15 times and an increase in tensile elongation by 6 to 940%. Therefore, the solvent-free molding of PPC using hot pressing at different temperatures without the need for additional plasticizers enables the production of materials with properties of both rigid plastics with a modulus of elasticity of 1.5–2.0 GPa and soft, pliable materials with a modulus around 200 MPa.

Additionally, thermomechanical analysis has revealed a significant increase in the volume of the sample by hundreds of percent in the range of 80—130 °C; this process is reversible at temperatures below 110 °C and partially reversible at 130 °C. According to TMA and a stress–strain test, PPC also exhibits large, reversible deformations that can be utilized in the creation of materials with shape memory.

In conclusion, we have proposed a new simple approach that allows for the soft tuning of the elastic properties of amorphous PPC containing predominantly carbonate units (>99%).

## Data Availability

The original contributions presented in this study are included in the article. Further inquiries can be directed to the corresponding authors.

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
