# Peer review of "Novel Simple Approach for Production of Elastic Poly(propylene carbonate)"

_polymers, 2024, doi:10.3390/polym16233248_

Round 1
Reviewer 1 Report (New Reviewer)
Comments and Suggestions for Authors
In this Article, the authors investigated the relationship between the mechanical properties of PPC films and hot pressing temperatures, along with the underlying mechanisms. The hot pressing process leads to the release of propylene carbonate, which lowers the glass transition temperature and alters the corresponding mechanical properties. However, it is confusing that the authors excluded the possibility of reduced molecular weight (MW) contributing to the lower Tg, even though the MW exceeded the entanglement molecular weight (Me). A combination of MW reduction and plasticization by propylene carbonate (PC) might have contributed to the observed results.
In addition, the following minor corrections should be addressed:
- Page 2, line 75: The authors should provide the full name of "PO" (propylene oxide) when it first appears in the text.
- Page 3, line 105: Reference 58 should be corrected to 59.
- Page 3, line 116: The term "BzF" should be explained for clarity.
- Page 3, line 122: The authors should clarify the solution composition “CH2Cl2/MeOH (10/1, v/v) solution in diethyl ether” or use the format “CH2Cl2/MeOH/Et2O (xx/xx/xx)” for consistency and better understanding.
Author Response
We would like to express our sincere gratitude to the reviewer for the helpful comments and suggestions. We have carefully revisited the article in accordance with all the comments. All the noticeable corrections are marked in red in the text.
Comment 1: In this Article, the authors investigated the relationship between the mechanical properties of PPC films and hot pressing temperatures, along with the underlying mechanisms. The hot pressing process leads to the release of propylene carbonate, which lowers the glass transition temperature and alters the corresponding mechanical properties. However, it is confusing that the authors excluded the possibility of reduced molecular weight (MW) contributing to the lower Tg, even though the MW exceeded the entanglement molecular weight (Me). A combination of MW reduction and plasticization by propylene carbonate (PC) might have contributed to the observed results.
Response: Recently we have conducted additional experiments, which have confirmed our conclusion. They are the following. We have synthesized PPC of lower and higher Mw and have determined its Tg. In addition, we have purified PPC from propylene carbonate by its dissolution and precipitation. After that, we have determined its Tg. In both cases, the similar Tg was observed. This information is added to the text.
Comment 2: Page 2, line 75: The authors should provide the full name of "PO" (propylene oxide) when it first appears in the text.
Response: corrected
Comment 3: Page 3, line 105: Reference 58 should be corrected to 59.
Response: corrected
Comment 4: Page 3, line 116: The term "BzF" should be explained for clarity.
Response: corrected
Comment 5: Page 3, line 122: The authors should clarify the solution composition “CH2Cl2/MeOH (10/1, v/v) solution in diethyl ether” or use the format “CH2Cl2/MeOH/Et2O (xx/xx/xx)” for consistency and better understanding.
Response: replaced by CH2Cl2/MeOH (10/1, v/v) solution in tenfold excess of diethyl ether
Reviewer 2 Report (New Reviewer)
Comments and Suggestions for Authors
1. On line 20, add the abbreviation (PC) after propylene carbonate.
2. In line 165, add the structural units to which the chemical shifts were assigned. These are not carbonate units, as the text suggests.
3. Add the structural formula of PPC to Figure 2.
4. Line 232 and Figure 6: Highlight the discussed NMR signals in the figure and add the structural formula of propylene carbonate.
Author Response
Comment 1: On line 20, add the abbreviation (PC) after propylene carbonate.
Response: corrected
Comment 2: In line 165, add the structural units to which the chemical shifts were assigned. These are not carbonate units, as the text suggests.
Response: corrected
Comment 3: Add the structural formula of PPC to Figure 2.
Response: corrected
Comment 4: Line 232 and Figure 6: Highlight the discussed NMR signals in the figure and add the structural formula of propylene carbonate.
Response: corrected
Reviewer 3 Report (New Reviewer)
Comments and Suggestions for Authors
I can tell the research designing is solid, and the results are supportive, but before final publications, I would suggest consider some improvments.
The most significant of this work is the synthesis and polymerization. Therefore, I suggest the authors explain the mechanism, including the initiation and chain propagation. A diagram would be a good idea.
Also, I think it would be good to in vestigate the effects of reaction conditions on the Mn, Mw, PDI and mechanical strength.
In addition, since the polymer is elastic, I suggest the authors may consider some additional mechanical tests, such as tear strength, Mooney viscosity, etc.
Author Response
We would like to express our sincere gratitude to the reviewer for the helpful comments and suggestions. We have carefully revisited the article in accordance with all the comments. All the noticeable corrections are marked in red in the text.
I can tell the research designing is solid, and the results are supportive, but before final publications, I would suggest consider some improvements.
Comment 1: The most significant of this work is the synthesis and polymerization. Therefore, I suggest the authors explain the mechanism, including the initiation and chain propagation. A diagram would be a good idea.
Response: Recently we have published a paper, in which we have described in details the polymerization kinetics and mechanism (Rzhevskiy, S.A.; Shurupova, O.V.; Asachenko, A.F.; Plutalova, A.V.; Chernikova, E.V.; Beletskaya, I.P. The Role of Ligand Exchange in Salen Cobalt Complexes in the Alternating Copolymerization of Propylene Oxide and Carbon Dioxide. Int. J. Mol. Sci. 2024, 25, 10946. https://doi.org/10.3390/ijms252010946). According to the suggestion of the Reviewer, we have added some comments about living nature of the polymerization and the scheme of the polymerization mechanism (page 4). Corresponding reference to this paper is added also (ref.62).
Comment 2: Also, I think it would be good to investigate the effects of reaction conditions on the Mn, Mw, PDI and mechanical strength.
Response: We are thankful to the reviewer’s proposal. As it was mentioned above, we have already published the effects of reaction conditions on the Mn, Mw, and PDI. Concerning the deep study of mechanical characteristics, this item is under study now. We believe that it should be published separately as the aim of the present research is to show plasticizing effect of propylene carbonate formed due to the partial depolymerization of PPC.
Comment 3: In addition, since the polymer is elastic, I suggest the authors may consider some additional mechanical tests, such as tear strength, Mooney viscosity, etc.
Response: We are thankful to the reviewer’s proposal. This deep study is also planned, and this question is beyond the scope of this article.
Round 2
Reviewer 3 Report (New Reviewer)
Comments and Suggestions for Authors
The authors have significantly improved the quality, especially taking my comments into considerations. Overall I agree the final publications.
Author Response
We are thankful for fruitful comments of the reviewer, which helped us to improve the quality of the paper
This manuscript is a resubmission of an earlier submission. The following is a list of the peer review reports and author responses from that submission.
Round 1
Reviewer 1 Report
Comments and Suggestions for Authors
The authors investigated and proposed increasing the elastic properties of atactic polypropylene carbonate (PPC) by selecting a hot pressing temperature between 110 and 140 °C. They synthesized atactic PPC through the ring-opening copolymerization of (rac)-propylene oxide and CO2, using a racemic salen complex of Co(III). Their thermomechanical analysis revealed significant volume expansion in the 80–130 °C range, with reversibility below 110 °C and partial reversibility at 130 °C. The authors demonstrated successful synthesis and presented their results systematically, supported by various characterizations. Overall, the manuscript was well-written and thoroughly supported by the characterizations. I recommend accepting the manuscript; however, there are a few grammatical errors that the authors should address before publication.
Comments on the Quality of English LanguageMinor grammatical errors are there; authors may need to correct them before publication.
Author Response
Dear reviewer, we are grateful to you for your positive review on our manuscript. We have very carefully checked the text and revised mistakes.
Reviewer 2 Report
Comments and Suggestions for Authors
This manuscript describes a novel and simple method for the production of elastomeric polypropylene carbonates. The properties of the prepared PPC and its reversibility were investigated at different hot pressing temperatures, confirming the existence of an effect of different hot pressing temperatures on its mechanical properties and reversibility. In the reaction, the PC released from PPC as a plasticiser can modulate its properties. The approach presented in the manuscript is satisfactory, but the logic is problematic, so please think carefully before making changes. In addition, the images in the manuscript are not aesthetically pleasing. Therefore, minor revisions were necessary before publication
1. Lines 48-60 mainly describe the claim that the low mechanical properties of polypropylene carbonate (PPC) are controversial because its different microstructures give it a wide range of mechanical properties, but do not explain its thermal stability, please describe its thermal stability at the beginning.
2. Please standardise figure sizes throughout the text.
3. In section 3.2.1, please provide a more detailed explanation of the correlation between temperature and PPC characteristics; the current data cannot be relied upon as reliable evidence and the narrative is not logical.
4. In Table 2, is there a problem with the elongation at break of a hot pressed 110°C sample stretched at a rate of 50 mm min-1, if so change it, and conversely explain this condition.
5. In parts 348-351, the effect of hot pressing temperature on the properties of PPC is explained, but is there any effect of stretching rate on its reversibility, please explain this.
6. Please provide an explanation as to why the 120°C sample is absent from the gradient test, by including this additional section.
Comments on the Quality of English LanguageModerate editing of English language required.
Author Response
We would like to express our sincere gratitude to the reviewer for the helpful comments and suggestions. We have carefully revisited the article in accordance with all the comments. All the noticeable corrections are marked in red in the text. We have also revised English.
Comment 1: Lines 48-60 mainly describe the claim that the low mechanical properties of polypropylene carbonate (PPC) are controversial because its different microstructures give it a wide range of mechanical properties, but do not explain its thermal stability, please describe its thermal stability at the beginning.
Response: We have carefully revised introduction and discussed the literature data about thermal stability PPC and its mechanical behavior.
Comment 2: Please standardise figure sizes throughout the text.
Response: Corrected.
Comment 3: In section 3.2.1, please provide a more detailed explanation of the correlation between temperature and PPC characteristics; the current data cannot be relied upon as reliable evidence and the narrative is not logical.
Response: We completely agree with the reviewer's comments and have revised section 3.2.1 in order to make it clearer and more logical.
Comment 4: In Table 2, is there a problem with the elongation at break of a hot pressed 110°C sample stretched at a rate of 50 mm min-1, if so change it, and conversely explain this condition.
Response: There was a mistake in the capture to Fig. 6 in the numbering of the curves. Now it is corrected: Stress–strain curves for PPC films, obtained by hot pressing at 110 (a), 130 (b), and 140oC (c), at stretching rate 50 (1), 10 (2), and 2 mm min-1 (3). According to the principle of time-temperature superposition, the increase in stretching rate results in the embrittlement of the polymer. Therefore, Young’s modulus and tensile strength increases. Elongation at break should decrease with the increase in stretching rate. However, in the course of mechanical tests, the break of the sample can occur at lower values of deformations due to different factors.
Comment 5: In parts 348-351, the effect of hot pressing temperature on the properties of PPC is explained, but is there any effect of stretching rate on its reversibility, please explain this.
Response: Thank you for this comment. We have added the explanation, which is based on the principle of time-temperature superposition for polymers.
Comment 6. Please provide an explanation as to why the 120°C sample is absent from the gradient test, by including this additional section.
Response: The results obtained at 110, 130, and 140oC show a clear trend. As the temperature of hot pressing increases, the amount of PC produced and the elasticity of PPC increase. We assume that the amount of PC that can be released after hot pressing at 120oC is low enough to be detected by spectroscopy methods, so we limited our experiments to three temperatures. It is evident that the behavior of PPC after pressing at 120oC should be intermediate between that of PPC subjected to hot press at 110 and 130oC.
Reviewer 3 Report
Comments and Suggestions for Authors
The manuscript deals with the structure modification of the poly(propylene carbonate) (PPC). The authors believed the approach was novel and led to a more elastic polymer. Moreover, the enhanced elasticity of the PPC was caused by the formation of propylene carbonate (PC) acting as a plasticizer during the hot pressing at temperatures ranging from 110 to 140°C.
In my opinion, the conclusions were not sufficiently supported by the data and were incorrect. Indeed, the proposed approach for PPC property modification was pointless. A PC, which was formed during the hot pressing of PPC at 110-140°C, was a product of side reactions corresponding to the material degradation. The confirmation of this conclusion is the fact that the Tg and Mn decreased, as well as DI increased (Table 1).
An easier way to increase the amorphousness and elastic properties of the PPC was to copolymerize the appropriate molar amount of propylene oxide with e.g., ethylene/butylene oxide.
Considering poor scientific novelty and reliability I cannot recommend the publication of the manuscript in polymers.
Please, find additional comments/remarks below:
1. The nomenclature is incorrect – e.g., it should be “poly(propylene carbonate)”.
2. Tg should be rounded to a whole number.
3. How long the hot pressing have been carried out? The amorphous polymers were obtained after hot pressing, however, did you observe the crystallization after several hours/days? Did your results were repeatable? The additional WAXS/DMA measurements would be welcomed.
4. Did you study the content of carbonate bonds before and after the hot pressing?
5. Why tougher samples were obtained after hot pressing at 140 °C than at 130 °C (Figure 6)?
6. Dynamic rheological measurements (at temperatures above 100 °C) would be welcomed to understand the materials behavior.
Comments on the Quality of English Language
The manuscript demands an extensive language editing.
Author Response
We would like to express our sincere gratitude to the reviewer for the helpful comments and suggestions. We have carefully revisited the article in accordance with all the comments. All the noticeable corrections are marked in red in the text. The English editing has been done also by native speaker.
Comment 1: A PC, which was formed during the hot pressing of PPC at 110-140°C, was a product of side reactions corresponding to the material degradation. The confirmation of this conclusion is the fact that the Tg and Mn decreased, as well as DI increased (Table 1).
Response: We agree with the reviewer that partial degradation of the polymer occurs resulting in the decrease of the MW. However, in all the cases MW exceeds the the entanglement molecular weight several times, which is indicated in the text. Therefore, the decrease of Tg cannot be caused by decrease of Mn. Besides, the strength characteristics also are independent from the MW of the polymer. In contrast, elongation at break can decrease with the decrease in the MW. However, we observe its increase, which indicates the rise in the elasticity of the sample. These changes in mechanical behaviour of the PPC can be caused by plasticization only.
Comment 2: An easier way to increase the amorphousness and elastic properties of the PPC was to copolymerize the appropriate molar amount of propylene oxide with e.g., ethylene/butylene oxide.
Response: We agree with the reviewer that copolymerization of PO with ethylene oxide or butylene oxide can increase the elastic properties of PPC. However, this approach has a number of disadvantages. The reactivity ratios of PO/epoxide, epoxide/CO2 are unknown. Besides, complete conversion of epoxides in the copolymerization with CO2 are rarely achieved. Hence, it is difficult to provide the insertion of necessary amount of flexible carbonate units (ethylene or butylene carbonate) in the macromolecule. Additionally, the search of the appropriate initiator for both epoxides is required, which can provide high chemoselectivity for both epoxides (> 99 % of carbonate units). In contrast, the suggested approach is much more easier to realize in practice and it is reproducible.
Comment 3: The nomenclature is incorrect – e.g., it should be “poly(propylene carbonate)”.
Response: Corrected.
Comment 4: Tg should be rounded to a whole number.
Response: Corrected.
Comment 5: How long the hot pressing have been carried out? The amorphous polymers were obtained after hot pressing, however, did you observe the crystallization after several hours/days? Did your results were repeatable? The additional WAXS/DMA measurements would be welcomed.
Response: Hot pressing was carried out for 10 minutes. This information is given in section 2.2. We have not observed crystallization as atactic polymers are unable to crystalize and are amorphous. This conclusion is supported by DSC data. DSC was carried out within 1 – 2 weeks after hot pressing.
Comment 6: Did you study the content of carbonate bonds before and after the hot pressing?
Response: Yes, we have studied it by 1H spectroscopy, see please the Fig. S5 (ESI). We have added the corresponding sentence to the text.
Comment 7: Why tougher samples were obtained after hot pressing at 140 °C than at 130 °C (Figure 6)?
Response: There is probably some misunderstanding. According to the Fig. 6 and Table 2, the increase in the temperature of hot pressing results in the decrease of the strength properties of the polymer as we stated in the text.
Comment 8: Dynamic rheological measurements (at temperatures above 100 °C) would be welcomed to understand the materials behavior.
Response: Dynamic mechanical analysis is another useful method to study mechanical behavior of the polymers in the wide range of temperatures and frequencies and give information about glass transition, changes of G¢ and G² modules. However, we believe that mechanical tests carried out in the present work give enough information about material’s behavior. More detailed study of dynamic rheological behavior can be a part of another research.
Round 2
Reviewer 2 Report
Comments and Suggestions for Authors
The Reviewer thanks the authors for the manuscript revision. All issues have been taken into account. The quality of presentation of the experimental data is sufficiently high. As a result, this work deserves publication in Polymers.
Comments on the Quality of English LanguageMinor editing of English language required.
Reviewer 3 Report
Comments and Suggestions for Authors
The manuscript still demands extensive language corrections.
The responses to the remarks were unsatisfactory and too overall.
I could not find the comparison between mol% of carbonate groups before and after hot-pressing of PPC. The authors presented Figure S5 for PPC after hot-pressing and “Figure S2a for PPC (before hot-pressing)”. Unfortunately, the Figure S2a showed the 1H NMR spectrum of PC - see at signal at ca. 4,2 ppm (integral equals 2.00).
I cannot agree with the Authors that elongation at break decreases with Mn decreasing.
The plasticization effect resulting in PC formation was not proved sufficiently.
The conclusion was not sufficiently supported by the data and the manuscript should be rejected.
Comments on the Quality of English Language
The manuscript still demands extensive language corrections.